# Characterizing Vision-Language-Action Models across XPUs: Constraints and Acceleration for On-Robot Deployment

**Kaijun Zhou**[1] **Qiwei Chen**[1*] **Da Peng**[1*] **Zhiyang Li**[1] **Xijun Li**[1] **Jinyu Gu**[1 2 ✉]

## Abstract

Vision-Language-Action (VLA) models are promising for generalist robot control, but on-robot deployment is bottlenecked by real-time inference under tight cost and energy budgets. Most prior evaluations rely on desktop-grade GPUs, obscuring the trade-offs and opportunities offered by heterogeneous edge accelerators (GPUs/XPUs/NPUs). We present a systematic analysis for low-cost VLA deployment via model–hardware co-characterization. First, we build a cross-accelerator leaderboard and evaluate model–hardware pairs under **CET** (Cost, Energy, Time), showing that "right-sized" edge devices can be more cost-/energy-efficient than flagship GPUs while meeting control-rate constraints. Second, using in-depth profiling, we uncover a consistent two-phase inference pattern: a compute-bound VLM backbone followed by a memory-bound Action Expert, which induces phase-dependent underutilization and hardware inefficiency. Finally, guided by these insights, we propose **DP-Cache** and **V-AEFusion** to reduce diffusion redundancy and enable asynchronous pipeline parallelism, achieving up to $2.9\times$ speedup on GPUs and $6\times$ on edge NPUs with only marginal success degradation. The example leaderboard website is: `https://vla-leaderboard-01.vercel.app/`.

## 1. Introduction

The rapid progress of Large Language Models (LLMs) and Vision-Language Models (VLMs) has catalyzed *robotic foundation models*, particularly Vision-Language-Action (VLA) models (Sapkota et al., 2025). By coupling perception, language understanding, and action generation, VLAs promise to turn natural-language intent into physical behaviors. Yet, deploying VLAs beyond carefully controlled lab setups remains challenging: embodied intelligence is a closed-loop *observe–infer–act* process, and the inference stage must meet strict real-time requirements to avoid latency-induced stuttering, unsafe oscillations, or task failure (Brohan et al., 2023b; Fu et al., 2024).

A key bottleneck is that state-of-the-art VLAs are often evaluated on desktop-grade GPUs (e.g., NVIDIA RTX 4090) (Kim et al., 2024; Liu et al., 2025b; Pertsch et al., 2025; Ma et al., 2025). While these platforms enable rapid prototyping, they obscure a practical deployment question: *What compute platform is "right-sized" for VLA computation on robots?* For mobile and low-cost robots—even humanoid robots whose prices have dropped to as low as nearly $10,000 (e.g., Unitree G1)—the deployment envelope is constrained not only by end-to-end latency, but also by acquisition cost and energy consumption. In this setting, relying on a flagship GPU may lead to poor cost-efficiency, leaving a rapidly growing ecosystem of alternative accelerators—GPUs, NPUs, and emerging XPUs—largely unexplored.

Meanwhile, the algorithmic landscape of VLA models has evolved quickly. Early systems relied on imitation learning policies (e.g., ACT and Diffusion Policy) (Zhao et al., 2023; Chi et al., 2024), and more recent approaches increasingly adopt Transformer-based backbones and stronger action generators (Liu et al., 2025b; Team et al., 2024; Hou et al., 2025). Architecturally, the field is shifting from monolithic designs that directly autoregress action tokens (Brohan et al., 2023a; Kim et al., 2024) to dual-system pipelines that decouple high-level perception/reasoning from low-level, high-frequency control (e.g., Hi-robot, $\pi_0$, and Gr00t) (Shi et al., 2025; Black et al., 2026; NVIDIA et al., 2025). Despite these advances, *system-level understanding* of VLA inference on constrained hardware is still limited. Existing acceleration efforts often target a single model or a single high-end platform (Yu et al., 2025; Kim et al., 2025; Wang et al., 2025a; Ji et al., 2025; Ma et al., 2025), making it difficult for practitioners to answer two fundamental deployment questions: (i) which model-hardware combinations can satisfy a required control frequency under cost/energy constraints, and (ii) what bottlenecks actually dominate VLA inference in realistic end-to-end loops.

---

*Equal contribution [1]School of Computer Science, Shanghai Jiao Tong University [2]Shanghai Syslong Information Technology Co., Ltd.. Correspondence to: Jinyu Gu <gujinyu@sjtu.edu.cn>.

*Proceedings of the 43rd International Conference on Machine Learning*, Seoul, South Korea. PMLR 306, 2026. Copyright 2026 by the author(s).

This paper investigates **low-cost, on-robot VLA deployment** from a *model–hardware co-characterization* perspective. We build an experimental framework to evaluate representative VLA pipelines across heterogeneous accelerators, and we report a multi-dimensional *leaderboard* that quantifies latency, energy, and cost-efficiency. Our results show that when these constraints are considered jointly, the RTX 4090 is *not* invariably the best choice for deployment; in many scenarios, "right-sized" edge accelerators provide a better performance-to-cost operating point.

Beyond benchmarking, we seek to understand the computational characteristics and bottlenecks of VLA inference. Using Roofline analysis (Williams et al., 2009) and end-to-end profiling, we uncover a consistent *two-phase* inference pattern in mainstream VLA pipelines: a *compute-bound phase* dominated by the VLM backbone, followed by a *memory-bound phase* dominated by the action generation module (e.g., iterative diffusion or flow-matching-based Action Expert). This phase transition can induce substantial hardware inefficiency in practice—a single platform may be well-matched to one phase but underutilized in the other—which helps explain the gap between peak throughput specifications and achieved end-to-end control frequency.

Guided by these insights, we finally develop *latency optimizations* tailored to VLA computational structure. We investigate combinatorial acceleration strategies spanning redundancy reduction, architectural efficiency, speculative inference, and system-level pipelining. Concretely, we propose *DP-Cache* to reduce redundancy inside the iterative diffusion process and *V-AEFusion* to fuse and pipeline the outer synchronous workflow. Across platforms, our optimizations deliver substantial end-to-end speedups (e.g., $1.9\times$ with DP-Cache and $1.3\times$ with V-AEFusion, up to $2.9\times$ on GPUs and $6\times$ on edge NPUs), while maintaining task success with only marginal degradation.

In summary, our contributions are:

- **Heterogeneous VLA–XPU leaderboard for economical deployment.** We systematically benchmark representative VLA pipelines across heterogeneous accelerators and report end-to-end performance under practical constraints, enabling practitioners to select "right-sized" on-robot compute platforms.

- **Systematic profiling of VLA inference and its two-phase bottlenecks.** We characterize hardware utilization during end-to-end control and reveal a compute-bound VLM phase followed by a memory-bound Action Expert phase, explaining resource underutilization and deployment inefficiencies across hardware platforms.

- **Revealing latency optimizations for VLA control loops.** Based on our profiling insights, we develop

and evaluate optimization methods including hardware-aware acceleration, achieving significant end-to-end speedups across platforms.

## 2. Related Work

### 2.1. Vision-Language-Action Models

VLA models translate multimodal observations and language instructions into robot actions. Early systems relied on imitation learning from teleoperated demonstrations (e.g., RT-1, ACT, RT-2) (Brohan et al., 2023b; Zhao et al., 2023; Brohan et al., 2023a). OpenVLA fine-tunes a pre-trained *VLM Backbone* and predicts discretized action tokens that are decoded into continuous controls (Kim et al., 2024). In parallel, diffusion-based policies (e.g., Diffusion Policy, RDT) generate continuous actions via iterative sampling, improving action fidelity (Chi et al., 2024; Liu et al., 2025b).

Recent work increasingly hybridizes VLM and diffusion. $\pi_0$ couples a VLM backbone for semantic understanding with a separate *Action Expert* trained via flow matching to produce continuous motor commands; SmolVLA and $\pi_{0.5}$ follow similar designs (Black et al., 2026; Shukor et al., 2025; Intelligence et al., 2025). Hierarchical variants (e.g., Hi-robot, G0, Gr00t) further decouple high-level perception/reasoning from low-level, high-frequency action generation (Shi et al., 2025; Jiang et al., 2025; NVIDIA et al., 2025). Because end-to-end latency is jointly determined by model structure and hardware, we analyze the computational behavior of these components in Section 4.

### 2.2. VLA Acceleration Methods

Acceleration methods typically target either the VLM backbone (autoregressive decoding and attention-heavy compute) or the Action Expert (iterative generation). We summarize prior efforts into four families:

- **Redundancy reduction (caching, sparsity).** Reuse temporally redundant computation or skip unnecessary layers/activations (e.g., Cache-VLA, DeeR, EfficientVLA, SparseVLM) (Xu et al., 2025; Yue et al., 2024; Yang et al., 2025; Zhang et al., 2025).

- **Architectural efficiency (compression, distillation).** Lightweight backbones and distill diffusion policies to reduce the number of sampling steps (e.g., RoboMamba, TinyVLA, VQ-VLA, Consistency Models) (Liu et al., 2024; Wen et al., 2024; Wang et al., 2025b; Song et al., 2023).

- **Speculative inference.** Reduce generation overhead for VLM decoding and diffusion sampling (Spec-VLA and TS-DP) (Wang et al., 2025a; Li et al., 2025).

- **System-level pipelining.** Overlap model inference

with robot execution to reduce synchronous blocking in the observe–infer–act loop (Real-Time Chunking and SmolVLA) (Black et al., 2025; Shukor et al., 2025).

## 2.3. Heterogeneous Hardware Landscape

VLA research has largely relied on NVIDIA platforms, often using high-end consumer GPUs (e.g., RTX 4090) (Black et al., 2026; Kim et al., 2024; Shi et al., 2025; Liu et al., 2025b; NVIDIA et al., 2025; Pertsch et al., 2025; Ma et al., 2025). However, on-robot deployment is also constrained by cost and power; a single flagship GPU can be poorly matched to end-to-end VLA pipelines, leading to underutilization. At the same time, edge inference hardware is diversifying beyond discrete GPUs, including embedded GPUs and accelerators such as *Huawei Ascend NPUs* (310B/310P) and *Intel XPUs* (Arc B60 Pro) [1]. These platforms differ in compute throughput, memory capacity, power consumption, and software stack, motivating a systematic comparison and selection mechanism.

## 3. Model-Hardware Pairing

VLA deployment couples heterogeneous model families with heterogeneous robot accelerators, making *model-hardware pairing* a practical bottleneck. The same model can run in real time on a desktop GPU yet fail to meet control-rate constraints on embedded SoCs, and model size alone is not a reliable indicator of latency. We therefore present a qualitative tiered view to build intuition, and a quantitative *Leaderboard* to ground deployment decisions in measured latency, cost, and power.

### 3.1. Model and Hardware Tier Classification

We first summarize the landscape with a tiered Consumer (model)–Producer (accelerator) matrix (Figure 1).

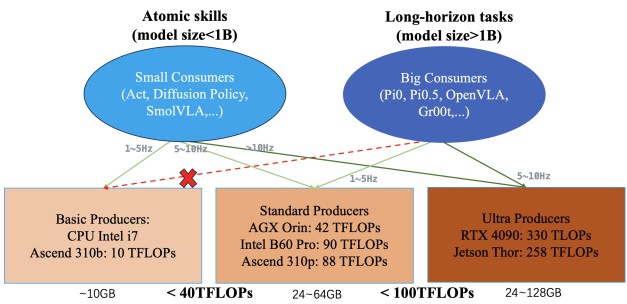

*Figure 1.* The diagram illustrates the compatibility matrix between VLA model types (Consumers) and hardware platforms (Producers), together with the hardware memory and peak FLOPs.

---

[1]Beyond NVIDIA GPUs, we currently choose hardware platforms from two major vendors for evaluation, Intel (one of the most impormant CPU vendors) and Huawei (one of the leading NPU vendors).

**Model tiers (Consumers).** We classify VLA models into two tiers using a 1B-parameter threshold: *Small Consumers* (e.g., ACT, Diffusion Policy, SmolVLA) and *Big Consumers* (e.g., Gr00t, $\pi_0$, $\pi_{0.5}$, OpenVLA). Small Consumers are typically efficient for atomic skills; Big Consumers offer stronger long-horizon capability but demand higher compute/memory bandwidth.

**Hardware tiers (Producers).** We categorize hardware platforms into three tiers based on their theoretical peak FP16/BF16 computing capability (FLOPS): *Basic Producers* (e.g., 11th Gen Intel i7-11700 CPU, Ascend 310B), *Standard Producers* (e.g., Intel B60 Pro, AGX Orin, Ascend 310P), and *Ultra Producers* (e.g., Jetson Thor, RTX 4090).

This clustering gives a fast feasibility prior based on two hard constraints—*compute* (FLOPS for the control rate) and *memory capacity* (VRAM to host model weights and activations): *Big Consumers* typically require *Standard/Ultra Producers* to meet both the real-time control rate and the VRAM budget, while *Basic Producers* are usually limited to *Small Consumers*. For example, the 7B OpenVLA cannot fit on the 12 GB Ascend 310B; although large models can be forced onto a commodity CPU via swap memory, the resulting disk I/O stalls drop inference to $< 1$ Hz, making it practically unusable. We therefore treat VRAM capacity as a hard feasibility constraint alongside the latency gate.

*Table 1.* $\pi_0$-related data presented in the Leaderboard.

| Model | Score | XPU Type | Memory(GB) | Bandwidth(GB/s) | Cost($) | Energy(kJ) | Time(ms) |
|-------|-------|----------|-----------|-----------------|---------|-----------|----------|
|       |       | 4090     | 24        | 1000            | 3500    | 2.398     | 102.3    |
|       |       | Thor     | 128       | 273             | 3400    | 1.282     | 246.0    |
| Pi0   | 86.0  | Orin     | 64        | 204             | 1999    | 1.866     | 920.6    |
|       |       | B60      | 24        | 456             | 599     | 6.363     | 306.5    |
|       |       | 310p     | 48        | 204.8           | 1030    | 2.618     | 818.0    |

### 3.2. Leaderboard for Model–Hardware Pairing

While tiering provides intuition, we further maintain an interactive *Leaderboard* that reports end-to-end inference latency across (model, XPU) pairs and augments latency with two on-robot constraints: *cost* (one accelerator per robot at scale) and *energy* (battery budgets, especially important for mobile robots). We summarize these constraints with **CET**: **T**ime (latency or control rate) as the feasibility gate, and **C**ost and **E**nergy as the primary deployment trade-offs among feasible pairs.

Table 1 shows a partial view of the *Leaderboard*, comparing $\pi_0$ performance across XPU types. For the model score, we use the average success rate across four LIBERO (Liu et al., 2023) subtasks (Spatial, Object, Goal, Long) to reflect model capability, which is hardware-independent. For a given VLA model, its capability (i.e., task success rate) is not affected by the underlying platform when the same precision (e.g., BF16) is used. For energy consumption, we measure it during inference on each platform using a physical power meter. This leaderboard makes pairing actionable:

*Figure 2.* VLA model performance across hardware accelerators. The bar chart displays inference frequencies (log scale) across hardware platforms for VLA models. The pink line represents capability scores on LIBERO (Liu et al., 2023). Horizontal lines indicate frequency thresholds (1Hz-30Hz).

it exposes when "smaller" models are slower (e.g., due to iterative diffusion steps) and when non-flagship accelerators are the better fit under cost/energy limits.

Figure 2 summarizes control-rate feasibility across heterogeneous platforms. Pairs below the frequency threshold are infeasible regardless of cost or power efficiency, so we apply latency-first screening before CET-based ranking.

Pairs combining Basic Producers (CPU and Ascend 310B) with Big Consumers are omitted from the figure, as they are pre-filtered by the VRAM feasibility constraint discussed above and thus excluded from CET ranking.

> **Finding #1. Model size is not a definitive predictor of inference frequency.**

Counter-intuitively, models with fewer parameters do not always yield higher inference speeds. For instance, the Small Consumer (Diffusion Policy) exhibits a lower inference frequency than the larger Big Consumer ($\pi_0$). This discrepancy is primarily driven by the *iterative nature* of the diffusion action expert. The Big Consumer Gr00t requires only 4 denoising steps, while the Small Consumer must execute 100 denoising steps sequentially. This counter-intuitive phenomenon caused by the denoising process also guides the design of our acceleration methods in Section 2.2.

Latency-first screening is necessary but insufficient for on-robot deployment; among feasible pairs, users must trade off *cost* and *energy*. Figure 3 gives a representative comparison of three VLA workloads ($\pi_0$, SmolVLA, and Gr00t) across platforms from the Standard to the Ultra tier.

> **Finding #2. Multi-dimensional analysis reveals that optimal hardware selection varies across VLA models and metrics.**

When users select CE or CET metrics, as shown in the bottom panel of Figure 4, the analysis is followed:

**CE Sorted (Cost-Energy Priority):** When latency is not a critical factor and the focus is instead on task completion and operational expenditure (hardware cost and electricity), users should prioritize cheaper, energy-efficient hardware. Here, the *Orin* is optimal for small consumers like SmolVLA, whereas the *310P* proves superior for heavier workloads like Gr00t and $\pi_0$.

**CET Sorted (All-Round):** When all three factors are balanced, the optimal hardware choice diverges: *Thor* for SmolVLA, *310P* for Gr00t, and *4090* for $\pi_0$. Notably, energy intrinsically includes a time component ($E = P \times t$), so CET naturally penalizes solutions that are either slow or power-hungry.

**Guideline for Deployment Platform Selection.** Building on the CET analysis above, we distill the Leaderboard into a concrete three-step selection guideline. Given a target model, users (i) directly select it in our interactive Leaderboard to retrieve profiling results across all tested hardware, (ii) apply *feasibility filters* that exclude platforms failing either the *VRAM capacity* check (OOM at load time) or the required *control frequency*, and (iii) sort the remaining feasible pairs by the priority metric aligned with their deployment scenario. For example, given $\pi_0$, the Leaderboard yields: *latency priority* (dynamic tasks) selects the RTX 4090 (fastest), with Jetson Thor as an alternative; *cost priority* (scale deployment) selects the Intel B60 Pro (lowest cost among feasible platforms), with Ascend 310P as an alternative; and *energy priority* (battery-powered robots) selects Jetson Thor, followed by AGX Orin. These recommendations follow directly from the CET analysis without any additional tuning, demonstrating that the Leaderboard turns model-hardware pairing from an intuition-driven process into a reproducible selection procedure.

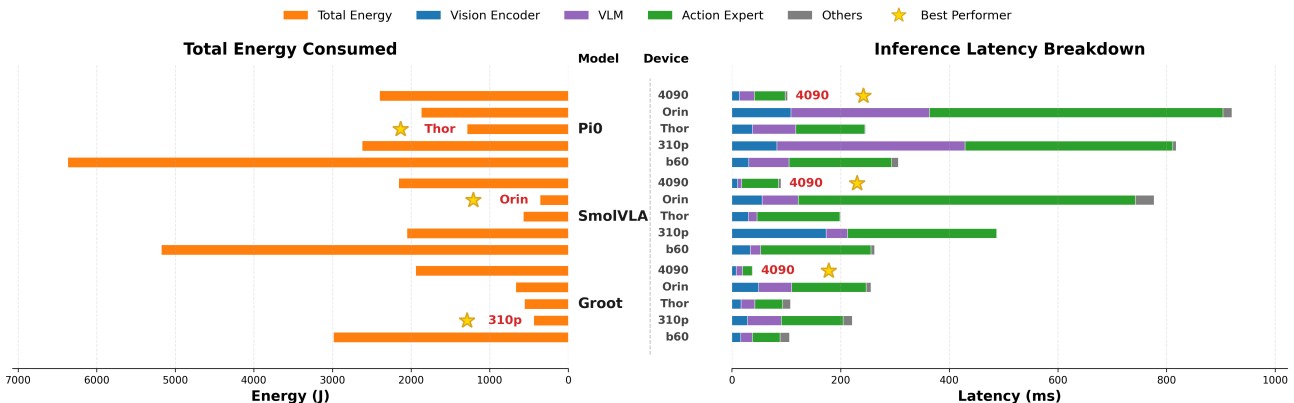

*Figure 3.* Energy-Latency Analysis: Energy Consumption and Latency Breakdown across Models and Hardware Platforms.

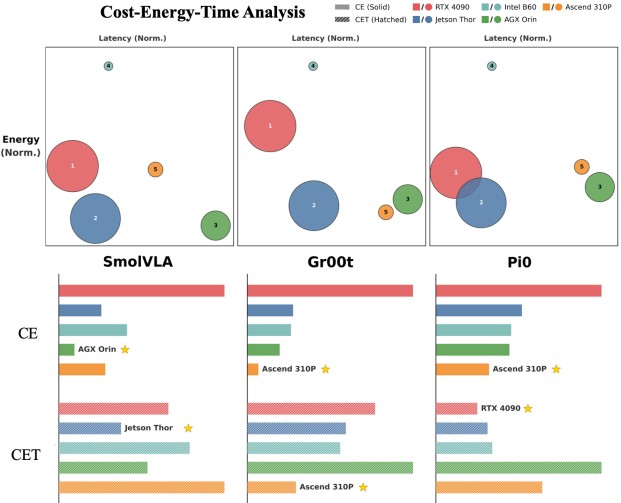

*Figure 4.* Hardware selection guidelines under multi-dimensional metrics (CET). The bubble size encodes hardware cost (Cost), the x-axis represents normalized inference latency (Time), and the y-axis represents energy consumption (Energy).

We further apply this guideline to $\pi_{0.5}$ and validate it via real-world deployment on a Franka arm in Appendix A.2, and we will keep updating the open-sourced *Leaderboard*.

## 4. VLA Computation Characterization

As highlighted in Finding #1 (Section 3.2), the counter-intuitive observation that smaller models may exhibit higher latency necessitates a deeper investigation into VLA workload characteristics.

Peak FLOPS/TDP provide only loose upper bounds; real deployment is dictated by *measured* utilization and bottlenecks. We therefore characterize VLA execution with fine-grained profiling (utilization/latency breakdown) and roofline analysis (compute vs bandwidth limits), to identify which stage dominates latency and which hardware resource constrains it.

### 4.1. Dual-phase Computational Imbalance

We conducte fine-grained profiling of $\pi_0$ on NVIDIA RTX 4090, AGX Orin and Jetson Thor using Nsight Systems (NVIDIA Corporation, 2026).

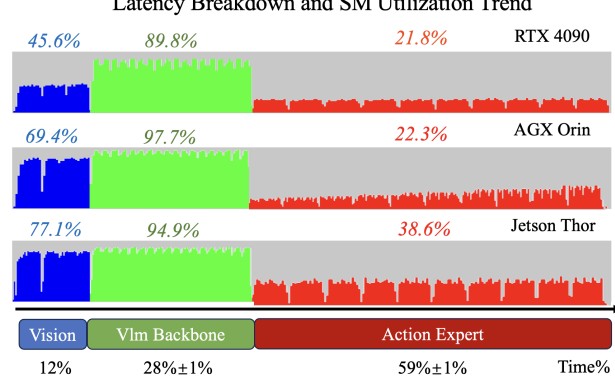

*Figure 5.* Profiling Results of $\pi_0$ on NVIDIA platforms. The blue, green, and red numbers on the profiling trace of each hardware correspond to the SM utilization of the same color phase.

Figure 5 compares latency breakdown across three stages (vision, VLM backbone, Action Expert) and corresponding SM utilization percentages. VLM backbone shows the highest SM utilization (typically above 90%) and Action Expert exhibits the lowest SM utilization (between 20% and 40%). This clearly demonstrates the two-stage characteristics of the VLA workflow.

> **Finding #3. VLA workflow exhibits a distinct dual-phase computational imbalance, leading to severe resource underutilization.**

For $\pi_0$, the VLM backbone achieves $3\times$ higher hardware utilization than the Action Expert, yet the Action Expert dominates latency ($2\times$ that of the VLM). On capable platforms like Standard and Ultra Producers, the serial dependency between the two phases forces high-performance compute units to idle during the inefficient Action Expert phase. In Section 5.3, we introduce **V-AEFusion**, a parallelization

strategy that pipelines the VLM and Action Expert, thereby mitigating this resource wastage and masking latency.

## 4.2. Computational Characteristics for Each Phase

Typically, autoregressive models (LLMs, VLMs) is memory-bandwidth bound due to intense KV-cache IO for decoding. Conversely, diffusion models for content generation (e.g., text-to-video (Zheng et al., 2024)) are compute-bound. This is because they process long input sequences (correlated with high resolution and frame counts (Wan et al., 2025)), resulting in quadratic computational complexity that saturates compute units.

To characterize the computational properties of the two phases in VLA workflow, we employ the roofline model (Williams et al., 2009), as shown in Figure 6. The roofline model relates throughput to operational intensity (FLOPs/Byte) and memory bandwidth; the ridge point (e.g., RTX 4090: 330 FLOPs/Byte) separates compute-bound from bandwidth-bound regimes.

> **Finding #4. VLA robotics inverts traditional generation bottlenecks: the VLM backbone is compute-bound, while the Action Expert is memory-bound.**

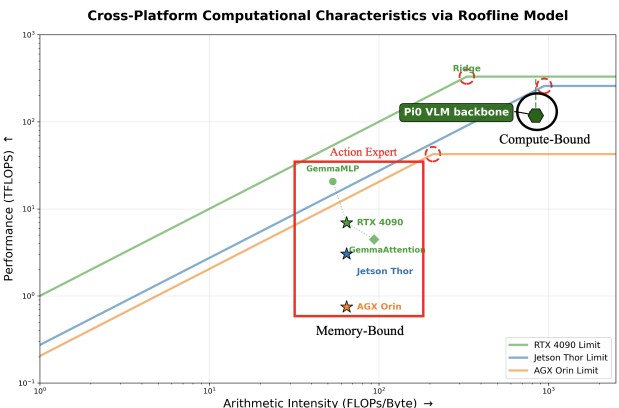

*Figure 6.* Roofline model analysis of $\pi_0$ computational characteristics. All three hardware platforms (RTX 4090, AGX Orin, Jetson Thor) test the GemmaDecoderlayer layer in the Action Expert (marked with asterisk). Additionally, RTX 4090 tests the VLM backbone, with separate profiling of GemmaMLP and GemmaAttention components.

**VLM Backbone (Compute-Bound):** The VLM backbone acts as a feature extractor, processing observations in a single forward pass (no autoregressive decoding).

The VLM backbone of $\pi_0$ comprises 18 DecoderLayer blocks. Each DecoderLayer forward pass requires 185.1G Flops of computation and accesses 110M parameters from memory, resulting in a total memory access of 220MB. For the VLM DecoderLayer, the computational intensity is calculated as $\frac{185.1\text{G Flops}}{220\text{MB}} \approx 840$ Flops/Byte. Since this inten-

sity is *above* the RTX 4090 ridge point (330 FLOPs/Byte), the layer is compute-bound on the RTX 4090.

**Action Expert (Memory-bound):** To satisfy control-rate requirements, the Action Expert uses relatively lightweight heads (e.g., 300M for $\pi_0$, 800M for Gr00t) yet still exhibits low SM utilization. Calculating the operational intensity ($I$) for the Action Expert on the RTX 4090 yields 64.5 FLOPs/Byte. Since this value falls significantly to the left of the hardware's ridge point, the workload is classified as memory-bound.

Notably, $I$ remains constant across platforms; thus on Jetson Thor and AGX Orin, $I = 64.5$ is far below their ridge points (945 and 208 FLOPs/Byte), confirming a universal bandwidth bottleneck.

In summary, VLA latency is constrained by distinct algorithmic characteristics across its two stages. The VLM backbone imposes substantial computational demands, creating a pronounced bottleneck on Basic and Standard platforms due to their limited arithmetic throughput. Conversely, the Action Expert phase is primarily constrained by memory bandwidth. Crucially, its inherent iterative serial dependency precludes effective parallel acceleration on hardware. Furthermore, on platforms with restricted memory bandwidth, this bottleneck is severely exacerbated by the iterative nature of diffusion policies (10–100 steps (Black et al., 2026; Chi et al., 2024)).

VLA optimization necessitates a multifaceted approach. We should explore different acceleration techniques for individual phases of the VLA models, and exploit a strategic architectural design to balance resource utilization across the two distinct computational stages. In the next section, we will delve into algorithmic optimizations and asynchronous execution flows designed to address this resource imbalance. Regarding bandwidth-oriented optimization, we identify it as a pivotal direction for future work.

## 5. Acceleration

### 5.1. Fusion of Orthogonal Acceleration Techniques

Here we use OpenVLA's workflow as an example, which as an autoregressive model, has a prefill phase and an iterative decode phase. Baseline OpenVLA require 6 decoding rounds to generate a 7-DoF action.

Given that the VLM backbone is primarily compute-bound, optimization focuses on reducing the arithmetic intensity per inference step. We choose Cache-VLA (Xu et al., 2025), which targets the prefill phase, and Spec-VLA (Wang et al., 2025a), which targets the decode phase. We additionally apply 4-bit quantization (Dettmers et al., 2023).

We adopt the default configurations from the original implementations for all acceleration methods. Table 2 presents

*Table 2.* Speedup and LIBERO Success Rate (SR) of OpenVLA under different acceleration methods.

| Method | Speedup | Freq. (Hz) | Spatial (%) | Object (%) | Goal (%) | Long (%) | Avg (%) |
|---|---|---|---|---|---|---|---|
| Baseline | 1.00× | 5.41 | 84.7 | 88.4 | 79.2 | 53.7 | 76.5 |
| Quant. (4-bit) | 1.14× | 6.17 | 77.4 | 64 | 75.8 | 53.4 | 67.7 |
| Cache | 1.16× | 6.29 | 83.8 | 85.8 | 76.4 | 52.8 | 74.7 |
| Spec. | 1.11× | 6.02 | 85.8 | 85.0 | 74.4 | 55.0 | 75.1 |
| Spec. + Cache | **1.29×** | **6.99** | 78.6 | 71.4 | 76 | 48 | 68.5 |
| Quant. + Cache | 1.27× | 6.84 | 76.6 | 62.2 | 72.4 | 49 | 65.1 |

the impact of these acceleration techniques on OpenVLA's end-to-end latency and task success rate.

The combination of Speculative and Cache achieves the highest speedup (1.29×), while incurring a 6.6% decrease in average success rate (SR) compared to Speculative alone. Notably, the speedup from Speculative relies heavily on the draft model's acceptance rate, making it highly sensitive to numerical precision during the verification phase. Quantization therefore undermines speculative acceleration, as its precision loss reduces the draft acceptance rate.

## 5.2. DP-Cache within Compiled Model

### 5.2.1. COMPILING

Model compiling is a common technique to optimize model inference performance. We leverage `torch.compile` to perform Just-In-Time (JIT) compilation, fusing PyTorch operations into optimized kernels to reduce overhead on NVIDIA GPUs. We utilize the Ascend Tensor Compiler (ATC) on Ascend NPUs to compile the full workflow into a static computation graph for efficient offline inference.

*Table 3.* Performance acceleration of $\pi_0$ using compilation across different hardware platforms.

| Hardware | Baseline (ms) | Compiled (ms) | Speedup (×) | Freq. (Hz) |
|---|---|---|---|---|
| RTX 4090 | 102.3 | 35.2 | **2.90×** | 28.41 |
| Jetson Thor | 246.0 | 163.0 | **1.51×** | 6.13 |
| Ascend 310P | 818.0 | 350.0 | **2.34×** | 2.86 |

We observe that the efficacy of model compilation exhibits significant hardware-dependent variability. For instance, the compiled $\pi_0$ model achieves a modest 1.51× speedup on Jetson Thor, whereas it attains a substantial 2.90× acceleration on the RTX 4090 and 2.34× on the Ascend 310P.

### 5.2.2. DP-CACHE: DIFFUSION POLICY CACHE

Reducing the number of iterations is critical for efficient Diffusion Policies (Chi et al., 2024; Hou et al., 2025). To address this, we implement the DP-Cache mechanism for the VLA action expert module. Drawing inspiration from strategies in video generation (Zhao et al., 2025; Liu et al., 2025a), DP-Cache accelerates inference by skipping redundant intermediate steps in the diffusion process.

To validate this approach, we conducted experiments on diffusion policies (shown in Figure 7). The upper plot reveals a stable segment where the relative L1 distance

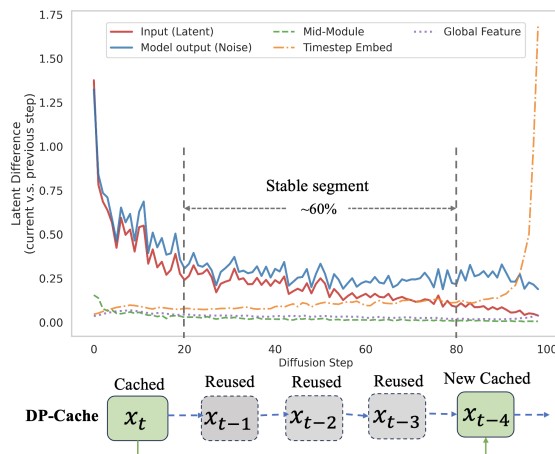

*Figure 7.* **Top:** Analysis of relative L1 differences ($L1_{rel}$) for key model components (model output, timestep embedding, noisy input, global feature, and mid-module blocks) between consecutive diffusion steps. **Bottom:** Schematic illustration of the DP-Cache mechanism. During the stable diffusion segment, computed results (green) are cached and broadcast to subsequent steps (gray).

($L1_{rel} = \frac{\|\mathbf{x}_{curr} - \mathbf{x}_{next}\|_1}{\|\mathbf{x}_{next}\|_1}$) between consecutive diffusion steps remains consistently low across model components. In our current implementation, the boundaries of this stable segment are fixed (approximately diffusion steps 20–80) and are determined via offline profiling of the diffusion trajectory prior to deployment. Leveraging this temporal redundancy, the lower schematic illustrates the DP-Cache mechanism, which broadcasts the cached result (green) to bypass computation for subsequent steps (gray) within this stable window, repeating this cycle throughout the stable segment.

*Table 4.* DP-Cache: Inference Latency and Capability (SR) on RTX 4090.

| Method | Speedup | Latency (ms) | Freq. (Hz) | Spatial (%) | Object (%) | Long (%) |
|---|---|---|---|---|---|---|
| Baseline (DP) | 1.00× | 378 | 2.6 | 75.4 | 52.4 | 74.1 |
| DP-Cache (S=4) | 1.89× | 200 | 5.0 | 78.9 | 51.4 | 76.9 |
| DP-Cache (S=8) | 2.09× | 181 | 5.6 | 71.6 | 46.1 | 73.4 |

Hyperparameter $S$ denotes the number of cache steps (i.e., $S = 4$ means cache the last 3 steps and recompute the 4-th step). Table 4 shows the impact of different cache steps on both inference latency and capability (SR). DP-Cache achieves 2.1× speedup with only a 3.6% SR drop compared to the baseline.

We further validate DP-Cache on $\pi_0$ in simulation (Appendix A.3) and on $\pi_{0.5}$ in real-world deployment.

*Table 5.* Franka real-world deployment: DP-Cache speedup and task success rate for $\pi_{0.5}$-droid (50 trials).

| Method | Speedup | Latency (ms) | SR (%) |
|---|---|---|---|
| Baseline ($\pi_{0.5}$) | 1× | 95 | 76 |
| DP-Cache (S=4) | 1.28× | 74 | 70 |

For the physical Franka robot on the table-cleaning task as shown in Figure 8, we observe a 1.28× speedup with a minor 6% SR drop, as summarized in Table 5.

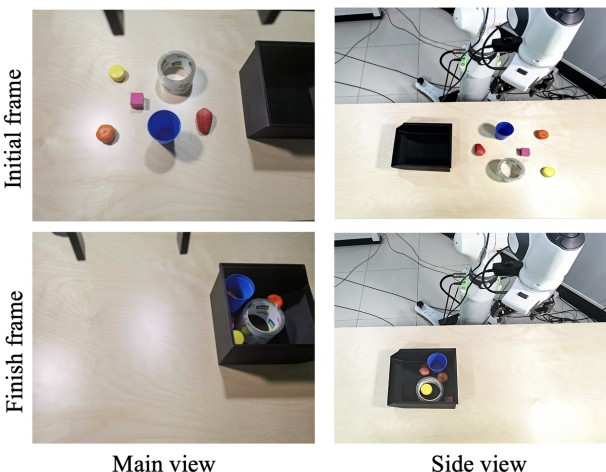

*Figure 8.* Visual demonstration of the manipulation task on Franka using $\pi_{0.5}$-droid: Cleaning up the table task.

*Table 6.* DP-Cache combined with model compiling across different hardware platforms (RTX 4090 and Ascend 310P).

| Hardware | Vanilla Latency (ms) | Optimized Latency (ms) | Speedup | Freq. (Hz) |
|---|---|---|---|---|
| 4090 | 378 | 129 | **2.9×** | 7.7 |
| 310P | 1853 | 310 | **6.0×** | 3.2 |

Table 6 shows the impact of combining optimization methods across different hardware platforms. On the Diffusion Policy model running on the Ascend 310P, DP-Cache combined with ATC compilation delivers a $6.0\times$ inference speedup, while ATC compilation alone provides a $3.5\times$ speedup ($1853\,\text{ms} \rightarrow 530\,\text{ms}$).

---

**Finding #5. Phase-aware, orthogonal optimizations must be composed with a global view.**

---

VLA latency is end-to-end, so effective acceleration requires jointly optimizing different phases (e.g., OpenVLA prefill vs. decode) and different levers (e.g., compilation reduces single-step overhead, while DP-Cache reduces diffusion iterations). More importantly, we need a unified framework that supports the integration of multiple optimization methods for VLA performance, analogous to xDiT (Fang et al., 2024), an acceleration framework for image and video generation.

### 5.3. V-AEFusion: Balancing the Low-Utilization Phase

#### 5.3.1. V-AEFUSION: VLM-ACTION EXPERT PIPELINE PARALLELISM

To further reduce end-to-end latency beyond chunk fusion, we exploit the temporal coherence of observations (Xu et al., 2025) and the computational imbalance between the VLM backbone and the Action Expert (Finding #3) to introduce *V-AEFusion*, a pipeline parallelism strategy illustrated in Figure 9.

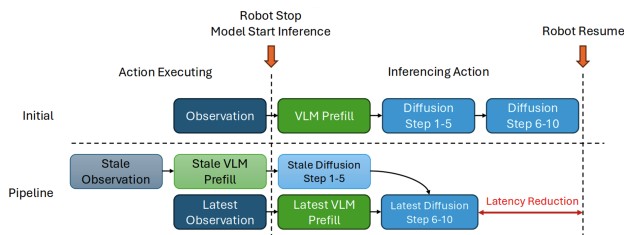

*Figure 9.* VLM-Action Expert Pipeline Parallelism based on Observation Similarity

**Pipeline Execution.** At timestep $t$, while the VLM processes the current observation ($Obs_t$), the Action Expert (AE) concurrently begins its early denoising steps using the *stale* KV cache from $Obs_{t-1}$. Once the VLM completes its forward pass, the AE switches to the *fresh* KV cache of $Obs_t$ for the remaining denoising steps, ensuring precise spatial refinement.

We refer to the denoising iterations that rely on previous-cycle visual features as *stale steps*. Two properties justify the use of stale features for the initial denoising steps without causing overshooting or degrading control stability:

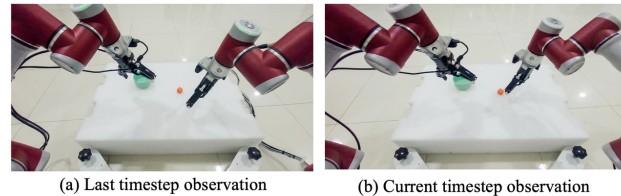

(a) Last timestep observation      (b) Current timestep observation

*Figure 10.* Observation inputs to the VLM at adjacent timesteps. Only the right distal end of the arm exhibits a small displacement.

**(1) High temporal coherence.** Under closed-loop control, the temporal gap between consecutive inference requests corresponds to the execution duration of a single action chunk. Within this brief window, the robotic arm's pose changes minimally, as shown in Figure 10. The VLM outputs exhibit high layer-wise cosine similarity between adjacent timesteps (Key States: $97.99\% \pm 0.8\%$, Value States: $96.91\% \pm 1.5\%$ across 200 consecutive inference cycles), confirming that stale features remain close approximations of the current observation.

**(2) Coarse-to-fine generation.** In diffusion models, early denoising steps establish the coarse action trajectory (low-frequency components), while later steps conditioned on fresh VLM features recover precise spatial details (high-frequency components). This late-stage refinement naturally corrects minor deviations introduced by stale conditioning (Song et al., 2023).

To validate V-AEFusion and determine the optimal number of stale denoising steps to overlap with the VLM phase, we evaluate $\pi_0$ on a JAKA S5 dual-arm robot performing two manipulation tasks (putting a block in a bowl and putting a cup in a bowl), as shown in Figure 11.

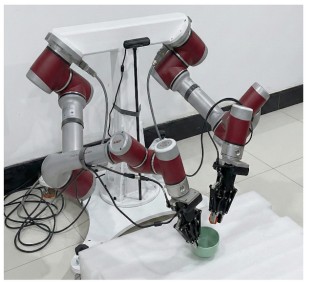 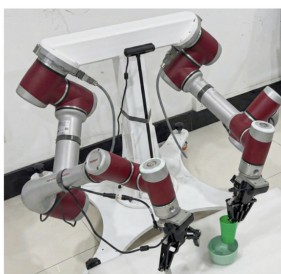

(a) Placing Block in Bowl     (b) Placing Cup in Bowl

*Figure 11.* Visual demonstration of two manipulation tasks on the JAKA S5 dual-arm robot using fine-tuned $\pi_0$.

**Ablation Study.** This design highlights a trade-off between speedup and performance fidelity: too few stale denoising steps yield limited overlap with the VLM, while too many cause conditioning drift and sharply reduce success rates. As shown in Table 7, performance degrades markedly once the number of stale steps exceeds six, consistent with the coarse-to-fine nature of diffusion generation. Since the VLM backbone and the Action Expert exhibit an approximate 1:2 latency ratio (Figure 5), we overlap half of the Action Expert's initial denoising steps (five stale steps) with the VLM phase. We further validate V-AEFusion on the LIBERO simulation benchmark as shown in Appendix A.4.

*Table 7.* Success rates (%) on two manipulation tasks as a function of the number of stale denoising steps (50 trials per task).

| Stale Steps | Putting Block in Bowl | Putting Cup in Bowl |
|---|---|---|
| 0 | 92 | 86 |
| 1 | 92 | 84 |
| 3 | 90 | 82 |
| **5** | **88** | **78** |
| 6 | 80 | 66 |
| 7 | 50 | 42 |
| 9 | 10 | 6 |

Although DP-Cache and V-AEFusion are currently deployed as training-free optimizations, they have the potential to achieve near-lossless acceleration when combined with training-based techniques such as consistency distillation (Song et al., 2023).

### 5.3.2. ACROSS HARDWARE

Since DP-Cache skips a fixed set of intermediate diffusion steps, its speedup remains consistent across different hardware platforms. V-AEFusion, on the other hand, tries to balance the computational utilization of the VLM and the Action Expert, so its speedup varies across hardware platforms.

> **Finding #6. Cross-hardware Impact: The same optimization method exhibits varying performance across platforms.**

Model compilation techniques exhibit varying effectiveness

*Table 8.* V-AEFusion Performance Comparison across Different Hardware Platforms.

| Method | 4090 (ms) | Orin (ms) | Thor (ms) | B60 (ms) | 310P (ms) |
|---|---|---|---|---|---|
| Baseline | 102 | 920 | 246 | 306 | 818 |
| V-AEFusion | 77 | 731 | 236 | 292 | 820 |
| Speedup | 1.32× | 1.26× | 1.04× | 1.05× | 1.0× |

across different hardware platforms for the same model. This variation stems from both the underlying hardware design and the compiler implementation within the software framework, as shown in Table 3. For asynchronous workflow methods like V-AEFusion, resource contention between parallelized model components must be considered. The aggregate resource demand of the VLM and the Action Expert can be mismatched with the hardware's compute-to-bandwidth balance, causing either bandwidth or compute bottlenecks and resulting in limited optimization benefits.

In summary, we characterize the VLA system by its *twofold serialization* nature: the sequential dependency between the VLM and the Action Expert (pipeline-level serialization), and the iterative denoising process within the Diffusion Action Expert (component-level serialization). As VLA models scale in parameter size, the Action Expert is poised to transition from a memory-bound to a compute-bound regime. Consequently, future architectural advancements must prioritize parallelism-centric designs (Rasley et al., 2020; Shoeybi et al., 2020) synergized with hardware-aware optimizations (Dao et al., 2022) to tackle challenging robotics tasks.

## 6. Conclusion

This paper studies *low-cost, on-robot* deployment of Vision-Language-Action (VLA) models via model–hardware co-characterization. We introduce a cross-accelerator **VLA-XPU Leaderboard** with **CET** (Cost, Energy, Time), and show that "right-sized" edge accelerators can be more cost- and energy-efficient than flagship GPUs under real-time constraints. We further identify a two-fold serialization nature of VLA inference: pipeline-level serialization between the VLM and the Action Expert, and component-level serialization within the diffusion process. This observation directly motivates **DP-Cache** and **V-AEFusion**, two training-free acceleration methods that deliver substantial end-to-end speedups across platforms with only marginal capability degradation.

**Limitations and Future Work.** Our leaderboard emphasizes efficiency, and our training-free optimizations may trade accuracy for latency. We will expand to broader benchmarks, models, and accelerators, and explore dedicated Action Expert optimizations as well as training-based methods (e.g., distillation/consistency) toward lossless acceleration.

## Acknowledgements

We sincerely thank the anonymous reviewers, whose reviews, feedback, and suggestions have significantly strengthened our work. We also thank Bin Xu from Shanghai Jiao Tong University for his assistance with the robot experiments. This work was supported in part by New Generation Information Technology Program from Shanghai Committee of Science and Technology (No. 25511104100), National Natural Science Foundation of China (No. 62432010), the Fundamental Research Funds for the Central Universities, and the Fundamental and Interdisciplinary Disciplines Breakthrough Plan of the Ministry of Education of China (JYB2025XDXM122). This work was also supported by OpenHarmony Embodied AI Project Management Committee (PMC).

## Impact Statement

This work contributes to the democratization of embodied AI by enabling high-performance VLA models to run on accessible, cost-effective edge hardware, lowering the economic barrier for robotics research and deployment. By addressing the critical trade-off between latency and energy consumption, our approach aligns with the goals of sustainable "Green AI," reducing the environmental footprint of large-scale robotic fleets. Furthermore, improving inference frequency directly enhances control stability, promoting safer human-robot interaction in dynamic real-world environments.

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

# A. Appendix

## A.1. Hardware Specifications

Our experimental setup comprises a heterogeneous set of devices, including a CPU (11th Gen Intel i7-11700), GPUs (RTX 4090, Jetson Thor, and AGX Orin), NPUs (Ascend 310B and 310P), and an XPU (Intel B60 Pro). All platforms support the PyTorch framework via the `torch.cuda`, `torch.xpu`, or `torch.npu` backends. Figure 2 summarizes the leaderboard results, and Table 9 lists the detailed hardware specifications used in our evaluation. [2]

*Table 9.* Hardware specifications for the GPU, XPU, and NPU platforms used in our evaluation.

| Hardware | BF16/FP16 | Memory | Memory Bandwidth |
|---|---|---|---|
| Ascend 310B | 10 TFLOP/s | 12 GB | 51.2 GB/s |
| Ascend 310P | 88 TFLOP/s | 48 GB | 204.8 GB/s |
| Intel B60 Pro | 90 TFLOP/s | 24 GB | 456 GB/s |
| NVIDIA AGX Orin | 42 TFLOP/s | 64 GB | 204 GB/s |
| NVIDIA RTX 4090 | 330 TFLOP/s | 24 GB | 1000 GB/s |
| NVIDIA Jetson Thor | 258 TFLOP/s | 128 GB | 273 GB/s |

## A.2. Real-world Energy Profiling for the Leaderboard

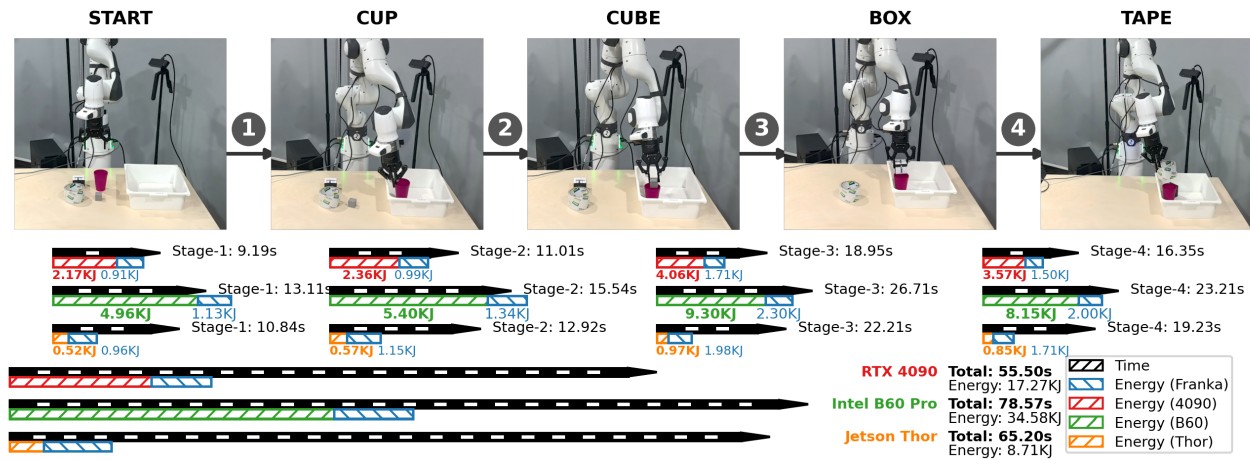

*Figure 12.* Real-world deployment: task execution stages with time and energy consumption analysis across hardware platforms on a Franka arm using $\pi_{0.5}$.

We independently measured the power consumption of both the inference hardware and the physical robot arm during execution, as shown in Figure 12. Among the platforms capable of inference at approximately 5 Hz (RTX 4090, Jetson Thor, and Intel B60 Pro), we find that Jetson Thor achieves the best end-to-end energy efficiency, while the RTX 4090 attains the shortest task completion time.

## A.3. DP-Cache Additional Evaluation

*Table 10.* DP-Cache: inference latency and task capability (success rate, SR) of $\pi_0$ on an RTX 4090.

| Method | Speedup | Latency (ms) | Spatial (%) | Object (%) | Goal (%) | Long (%) | Avg (%) |
|---|---|---|---|---|---|---|---|
| Baseline ($\pi_0$) | 1.00× | 102 | 64.0 | 78.4 | 72.8 | 35.2 | 62.6 |
| DP Cache(S=4) | 1.29× | 79 | 63.0 | 76.0 | 73.0 | 36.0 | 62.0 |

[2] Since the dense FP16/BF16 throughput of NVIDIA Jetson Thor and Intel B60 Pro is not officially reported, we estimate it as half of their INT8 TOPs following common vendor practice.

For $\pi_0$ in the LIBERO simulation on an RTX 4090, DP-Cache achieves a $1.29\times$ speedup with only a 0.6% absolute drop in average success rate, as shown in Table 10.

### A.4. V-AEFusion Additional Evaluation

We validate V-AEFusion with five stale steps on the LIBERO simulation benchmark as shown in Table 11. A stale-step count of zero corresponds to the vanilla (unoptimized) baseline. V-AEFusion achieves a $1.3\times$ speedup with only a 0.3% average SR drop on simulated LIBERO tasks and a 4% drop on real-robot pick-and-place tasks.

*Table 11.* V-AEFusion: inference latency and capability comparison of $\pi_0$ on RTX 4090.

| Method | Speedup | Latency (ms) | Spatial (%) | Object (%) | Goal (%) | Long (%) | Avg (%) |
|---|---|---|---|---|---|---|---|
| Baseline ($\pi_0$) | $1\times$ | 102 | 64.0 | 78.4 | 72.8 | 35.2 | 62.6 |
| V-AEFusion | $1.3\times$ | 77 | 60.2 | 76.6 | 77.4 | 35.0 | 62.3 |

