# OpenReview forum: "Characterizing Vision-Language-Action Models across XPUs: Constraints and Acceleration for On-Robot Deployment"
_ICML.cc/2026/Conference — ICML 2026 regular_

### Official Review · Reviewer_DyVD · 2026-02-26

**Soundness:** 3
**Presentation:** 3
**Significance:** 3
**Originality:** 3
**Overall Recommendation:** 4
**Confidence:** 3

**Summary:**

This paper presents a systematic analysis for low-cost VLA deployment through model-hardware co-characterization. It builds a cross-accelerator leaderboard for model-hardware pair evaluation. In addition, it proposes DP-Cache and V-AEFusion to reduce diffusion redundancy and enable asynchronous pipeline parallelism.

**Compliance With Llm Reviewing Policy:**

Affirmed.

**Final Justification:**

The authors have resolved my questions and I keep my ratings of weak accept.

**Key Questions For Authors:**

1. More theoretical insights could be provide to further improve the novelty of this paper.

**Limitations:**

yes

**Strengths And Weaknesses:**

Strengths:
1. This paper is well written and clear structured.
2. Empirical results demonstrate the effectiveness of the proposed method.

Weaknesses:
1. This paper mainly focus on architectural pipelines, more theoretical insights could further strengthen the novelty of this paper.

---

> ### Author Rebuttal · Authors · 2026-03-30
>
> We thank the reviewer for the positive assessment and the recognition that our paper is "well written and clearly structured" with empirical results that "demonstrate the effectiveness" of our methods. We respond to your comments as follows and sincerely hope that our rebuttal could properly address your concerns.
>
> ***Q1 & W1: This paper mainly focus on architectural pipelines, more theoretical insights could further strengthen the novelty of this paper.***
>
> We appreciate this suggestion and revise the paper to make theoretical insights more clear. We would like to highlight that our paper contributes both practical methods and tools and some theoretical insights:
>
> On the practical side, we contribute:
>
> 1. A **cross-accelerator VLA leaderboard** with CET metrics covering 7 models × 7 hardware platforms.
> 2. Identification of the **two-fold serialization nature** of VLA inference: pipeline-level serialization between VLM and Action Expert, and component-level serialization within the diffusion process, which directly motivates DP-Cache and V-AEFusion as training-free acceleration methods achieving up to 2.9× on GPUs and 6× on edge NPUs.
> 3. **Profiling tools and pipelines** across NVIDIA, Intel, and Huawei platforms (GPUs/XPUs/NPUs).
>
> On the theoretical side, our insights explain why these accelerations work:
>
> 1. **Roofline analysis** revealing the two-phase computational imbalance (e.g. $\pi_0$ VLM is compute-bound and Action Expert is memory-bound on RTX 4090), with fine-grained latency breakdown and SM utilization profiling showing that the VLM backbone achieves 3× higher utilization yet the Action Expert dominates latency at 2× (Findings #3–#4).
> 2. DP-Cache is grounded in the observation that the **diffusion trajectory exhibits a stable segment** (60% of steps) where $L1_{rel}$ between consecutive steps remains consistently low, meaning intermediate computations are temporally redundant and can be safely skipped.
> 3. V-AEFusion is justified by the **high temporal coherence** of robotic observations: VLM outputs between adjacent timesteps show high cosine similarity across layers (Key States: 97.99% ± 0.8%, Value States: 96.91% ± 1.5% across 200 consecutive inference cycles), making stale features reliable proxies for early denoising steps.

---

> > ### Author Rebuttal · Reviewer_DyVD · 2026-04-02
> >
> > The authors have resolved my concerns, thus I decide to keep my ratings of weak accept.

---

> > > ### Author Response · Authors · 2026-04-03
> > >
> > > Thank you for the positive feedback. We appreciate your valuable time and effort.

---

### Official Review · Reviewer_DqWH · 2026-03-05

**Soundness:** 3
**Presentation:** 3
**Significance:** 2
**Originality:** 3
**Overall Recommendation:** 4
**Confidence:** 2

**Summary:**

This paper addresses the gap between high-end GPU-based evaluations of Vision-Language-Action (VLA) models and the practical constraints of on-robot deployment (cost, energy, and latency). The authors introduce a systematic benchmarking framework across heterogeneous accelerators (GPUs, NPUs, XPUs) and propose a "CET" (Cost, Energy, Time) metric to identify "right-sized" hardware. Through detailed profiling, the paper reveals a consistent two-phase inference pattern in VLAs: a compute-bound VLM backbone followed by a memory-bound Action Expert. Based on these insights, the authors propose two optimization techniques: DP-Cache (to reduce redundancy in diffusion steps) and V-AEFusion (to pipeline the two phases). Experiments demonstrate significant speedups (up to 2.9x on GPUs and 6x on edge NPUs) with marginal performance degradation.

**Compliance With Llm Reviewing Policy:**

Affirmed.

**Key Questions For Authors:**

- Regarding V-AEFusion: Could you provide more technical details on how V-AEFusion handles the data dependency between the VLM backbone and the Action Expert? If they are pipelined asynchronously, does this introduce a lag between perception and action that might affect closed-loop control stability in highly dynamic environments?
- Baseline Quality: In Table 4, the baseline Diffusion Policy has a relatively low success rate on the "Goal" task (52.4%). How confident are you that the DP-Cache method would maintain performance on a stronger, fully converged baseline?
- Memory Constraints: The paper focuses on Latency, Cost, and Energy. However, memory capacity (VRAM) is often a hard constraint for edge devices. Did you encounter scenarios where "right-sized" hardware failed simply due to OOM (Out of Memory) errors, even if compute was sufficient?

**Limitations:**

yes

**Strengths And Weaknesses:**

Strengths:
- Systematic Benchmarking & Practical Relevance (Significance): The paper tackles a highly practical and overlooked problem in embodied AI: hardware selection under constraints. The introduction of the CET (Cost, Energy, Time) metric and the cross-platform leaderboard (NVIDIA, Intel, Huawei) provides valuable guidance for practitioners, moving beyond simple FLOPs comparisons.
- Insightful Profiling (Soundness): The Roofline analysis and breakdown of the "Two-Phase" bottleneck (Compute-bound VLM vs. Memory-bound Action Expert) are technically sound and offer a clear explanation for why smaller models (like Diffusion Policy) can sometimes be slower than larger ones due to iterative sampling. This insight is valuable for the community.
- Effective Optimization Strategies (Originality/Soundness): The proposed optimizations are well-motivated by the profiling results. DP-Cache effectively exploits temporal redundancy in diffusion steps, and the speedups reported (especially on edge devices like Ascend 310P) are impressive.
- Clear Presentation: The paper is well-structured. The distinction between "Consumer" (Models) and "Producer" (Hardware) tiers is an intuitive way to frame the compatibility problem.
Weaknesses:
- Limited Evaluation of "V-AEFusion" (Soundness): While the abstract and introduction mention V-AEFusion as a key contribution for pipelining, the main text (Section 5) focuses heavily on DP-Cache and compilation. The specific implementation details and ablation studies for V-AEFusion seem under-represented in the provided text compared to DP-Cache. It is unclear how the asynchronous pipelining handles the dependency between the VLM output and the Action Expert input.
- Baseline Comparisons (Soundness): In Section 5.2.2, the authors mention training a new Diffusion Policy baseline because OpenVLA's baseline wasn't available. However, the success rate of their baseline on the "Goal" task is notably low (52.4%). This raises a concern: are the speedups obtained on a model that is already suboptimal? Does the degradation from DP-Cache make the model unusable for harder tasks?

---

> ### Author Rebuttal · Authors · 2026-03-30
>
> We thank the reviewer for the insightful and valuable comments. We especially appreciate the recognition that our work tackles "a practical and overlooked problem," our Roofline analysis is "technically sound," DP-Cache speedups are "impressive," and the Consumer-Producer framing is "an intuitive way to frame the compatibility problem." We respond to your comments as follows and sincerely hope that our rebuttal properly addresses your concerns.
>
> ***Q1 & W1. V-AEFusion: How does it handle data dependency between VLM and Action Expert? Does asynchronous pipelining introduce perceptual lag affecting closed-loop stability in dynamic environments?***
>
> We provide a detailed technical explanation of the dependency handling with newly added evaluation using π₀ on a JAKA S5 dual-arm robot (RTX 4090, 50 trials) as shown in https://re-03-figures.vercel.app/ Figure 1.
>
> **Pipeline Execution**: At timestep t, while the VLM processes current observation ($Obs_t$), the Action Expert (AE) concurrently begins its early denoising steps using the stale KV cache from $Obs_{t-1}$. Once the VLM completes its forward pass, the AE switches to the fresh KV cache of $Obs_t$ for its final denoising steps to ensure precise spatial refinement.
>
> Using stale features for the initial steps may cause minor performance drops but does not cause overshooting or affect control stability. Stability is supported by:
>
> 1. **High temporal coherence**: Under closed-loop control, the temporal gap between consecutive requests corresponds to the execution duration of a single action chunk. Within this brief window, the robotic arm's pose changes minimally (https://re-03-figures.vercel.app/ Figure 2). The VLM outputs exhibit high layer-wise cosine similarity between adjacent timesteps (Key States: 97.99% ± 0.8%, Value States: 96.91% ± 1.5% across 200 consecutive inference cycles), confirming that stale features remain close approximations of the current observation.
> 2. **Coarse-to-fine generation**: In diffusion models, early denoising steps establish the rough action trajectory (low-frequency components), while the final steps conditioned on fresh VLM features recover precise spatial details. This late-stage refinement corrects minor deviations.
>
> "Stale Steps" indicates how many denoising steps use features from the previous VLM cycle. V-AEFusion using up to 5 stale steps maintains strong task performance with only 4–8% SR drop as shown below, while reusing more steps causes sharp degradation. We will include this ablation in the revised paper.
> |Stale Steps| Putting Block in Bowl (SR%)| Putting Cup in Bowl (SR%)|
> |:---:|:---:|:---:|
> |0 (Baseline)|92|86|
> |3|90|82|
> |5 (V-AEFusion)|88|78|
> |7|50|42|
>
> For highly dynamic environments (e.g., over 50 Hz), we agree this may remain a limitation. Selectively recomputing observation patches with significant motion can be our future work.
>
> ***Q2 & W2. Table 4 baseline has low Goal success (52.4%) — would DP-Cache maintain performance on a stronger, fully converged baseline?***
>
> To address this concern, we added a new DP-Cache evaluation on π₀ in simulation and π₀.₅ on a physical robot (https://re-03-figures.vercel.app/ Figure 3), both are strong, fully converged models.
>
> For π₀ in LIBERO simulation on RTX 4090, DP-Cache achieves 1.29× speedup with only ~0.6% absolute drop in average success rate as shown below:
> ||Speedup|Latency (ms)|Spatial (%)|Object(%)|Goal (%)|Long (%)|Avg (%)|
> |:-:|:-:|:-:|:-:|:-:|:-:|:-:|:-:|
> |Baseline($\pi_0$)|1x|102|64.0|78.4|72.8|35.2|62.6%|
> |DP-Cache(S=4)|1.29x|79|63.0|76.0|73.0|36.0|62.0%|
>
> For the physical robot with π₀.₅ of cleaning table task (Franka, RTX 4090, 50 trials), we observe 1.28× speedup with a minor 6% SR drop as shown below:
> |Franka|Speedup|Inference Latency (ms)|Task SR (%)|
> |:-:|:-:|:-:|:-:|
> |Baseline($\pi_{0.5}$-droid)|1x|95|76|
> |DP-Cache(S=4)|1.28x|74|70|
>
> We will add DP-Cache evaluation on π₀ and π₀.₅ in the revised version.
>
> ***Q3. Memory capacity (VRAM) is often a hard constraint for edge devices. Did you encounter OOM scenarios where "right-sized" hardware failed despite sufficient compute?***
>
> Memory capacity is indeed a constraint we considered, though it was not explicitly highlighted in the paper. In our framework, hardware classified as "right-sized" for "Big Consumers" (i.e., Standard or Ultra Producers) inherently possesses sufficient VRAM to run these models.
>
> Pairing Big Consumers with Basic Producers directly leads to OOM errors. This is reflected in Figure 2, where bar charts for Big Consumers exclude the CPU and NPU 310B. These pairs are removed from the selection space entirely. For instance, the 7B OpenVLA cannot fit on the 12GB Ascend 310B. While we could technically force large models onto an Intel i7 CPU by leveraging swap memory, the resulting disk I/O latency makes inference practically unusable (< 1Hz).
>
> In the revised manuscript, we will explicitly incorporate memory capacity as a hard feasibility constraint within our tier classification.

---

> > ### Author Rebuttal · Reviewer_DqWH · 2026-04-02
> >
> > Thank you for the thoughtful rebuttal. I am satisfied that my main concerns have been addressed.

---

> > > ### Author Response · Authors · 2026-04-03
> > >
> > > Thank you for the positive feedback. We appreciate your valuable time and effort.

---

### Official Review · Reviewer_ZNEF · 2026-03-13

**Soundness:** 3
**Presentation:** 2
**Significance:** 3
**Originality:** 3
**Overall Recommendation:** 4
**Confidence:** 2

**Summary:**

This paper studies the deployment of VLA models on heterogeneous hardware platforms for real-time robotic control. The authors build a cross-accelerator leaderboard evaluating model–hardware pairs under a CET (Cost–Energy–Time) metric and analyze VLA inference behavior using profiling and roofline analysis. They identify a consistent two-phase computation pattern where a VLM backbone is compute bounded and the Action Expert is memory bounded. The authors then propose two acceleration techniques, DP-Cache for reducing redundant diffusion steps and V-AEFusion for pipelining the VLM and action expert stages. These optimizations improve inference speed across hardware platforms while maintaining similar task success rates.

**Compliance With Llm Reviewing Policy:**

Affirmed.

**Final Justification:**

The authors have resolved my concerns in rebuttals and I decide to keep my ratings of weak accept.

**Key Questions For Authors:**

1. From the CET evaluation results, is there a clear guideline on how to select a deployment platform for a given model?
2. DP cache is only applicable to diffusion based action expert. How would you increase the efficiency for autoregressive action expert?

**Limitations:**

yes

**Strengths And Weaknesses:**

Strengths
1. The motivation is clear. Deploying VLA models on resource-constrained robotic platforms is a practical problem.
2. The proposed cross-accelerator leaderboard provides a useful framework for evaluating model–hardware combinations under multiple constraints.
3. The identification of the two-phase computational structure is interesting.

Weaknesses
1. While the system analysis is interesting, the proposed optimizations lack comparison with other baselines focusing on VLA efficiency. In addition, it seems some acceleration methods reduce inference latency at the cost of slight performance degradation as shown by Table 2.
2. The experiments focus primarily on LIBERO tasks, which is known to be saturated. Generalizing to other robotic benchmarks or more complex manipulation scenarios would be more interesting.
3. Although the paper discusses on-robot deployment, most of the evaluation appears to be conducted in simulation or controlled hardware experiments. Demonstrating consistent results on physical robots would strengthen the claims.
4. The leaderboard and CET evaluation framework are useful but there is no clear conclusion on how to select a deployment platform achieving the best trade-off.

---

> ### Author Rebuttal · Authors · 2026-03-30
>
> We thank the reviewer for the insightful and valuable comments. We especially appreciate the recognition that our motivation is "clear" and "practical," our cross-accelerator leaderboard provides "a useful framework," and the two-phase computational identification is "interesting." We respond to your comments as follows and sincerely hope that our rebuttal properly addresses your concerns.
>
> ***Q1 & W4. From the CET results, is there a clear guideline on how to select a deployment platform for a given model?***
>
> Yes. Given a target model, users can directly select it in our interactive leaderboard and view profiling results across all tested hardware. After applying a latency feasibility filter (excluding platforms below the required control frequency), users sort by their priority metric to identify the best platform.
>
> For example, given π₀:
> - **Latency priority** (dynamic tasks): RTX 4090 (fastest), alternative: Jetson Thor
> - **Cost priority** (scale deployment): Intel B60 Pro (lowest cost among feasible), alternative: Ascend 310P
> - **Energy efficiency priority** (battery-powered robots): Jetson Thor, followed by AGX Orin
>
> These recommendations come directly from the CET analysis. We will add a concise "How to Use" guide with such worked examples in the revised version.
>
> ***Q2. DP-Cache only applies to diffusion-based Action Experts — how would you increase efficiency for autoregressive Action Experts?***
>
> As demonstrated in Section 5.1 and Table 2, we have evaluated efficiency optimizations specifically targeted at the autoregressive Action Expert OpenVLA. We utilize Speculative Decoding to accelerate the decode phase, KV-cache optimization (Cache) for the prefill phase, and 4-bit Quantization to reduce the memory footprint. To maximize acceleration, we test combinations of these orthogonal methods. The key takeaway is that these methods are complementary but require careful composition (e.g., Quantization + Cache degrades to 65.1% vs. 76.5% baseline). This motivates our Finding #5: phase-aware optimizations necessitate a global view.
>
> ***W1. No comparison with other baselines focusing on VLA efficiency. Some acceleration methods degrade performance (Table 2).***
>
> 1. Baseline Comparisons: We respectfully note that our paper does compare against relevant methods. Table 2 evaluates Cache (Cache-VLA), Spec. (Spec-VLA), and Quant. (Qlora) for autoregressive VLAs, which represent the main families of existing acceleration baselines in this domain.
> 2. Performance Degradation: This is a critical trade-off that must be carefully considered during real-world deployment. In fact, quantifying the exact trade-offs of different acceleration methods is one of the main contributions of this paper. Exploring lossless acceleration could be a direction for our future work.
>
> ***W2 & W3. Experiments focus on LIBERO, which is known to be saturated. More complex benchmarks would be more convincing. Most evaluation is in simulation or controlled hardware — real-robot results would strengthen claims.***
>
> Thanks for your meaningful suggestion. We agree that broader evaluation strengthens our claims. A real-robot CET deployment analysis validates our leaderboard's practical utility. Our acceleration methods (DP-Cache and V-AEFusion) exploit intrinsic computational properties of the diffusion process, so the speedups generalize across workloads. To verify that performance is preserved on more challenging settings, we have added two sets of real-world physical robot experiments:
>
> 1. **Real-world energy profiling for the leaderboard**: We independently measured power consumption for both the inference hardware and the physical robot arm execution, as shown in https://re-02-figures.vercel.app/ Figure 1. Targeting platforms capable of ≥5 Hz inference (RTX 4090, Jetson Thor, and Intel B60 Pro), we found that **Jetson Thor** demonstrates the most optimal end-to-end energy efficiency, while **RTX 4090** delivers the fastest task completion time.
>
> 2. **DP-Cache and V-AEFusion in real-world settings**: To demonstrate that our optimizations generalize to complex tasks and real robots, we newly added evaluations on π₀ and π₀.₅ as shown below (on RTX 4090, 50 trials). For π₀ on a physical JAKA S5 robot, V-AEFusion achieves a ~1.3× speedup with a minor SR drop of 4–8% on Pick-and-Place tasks (https://re-02-figures.vercel.app/ Figure 2). For π₀.₅ on a Franka arm, DP-Cache achieves 1.28× speedup with marginal success degradation on Cleaning Table task (https://re-02-figures.vercel.app/ Figure 3).
>
> |JAKA S5|Speedup|Latency (ms)|Putting Block in Bowl (SR%)|Putting Cup in Bowl (SR%)|
> |:-:|:-:|:-:|:-:|:-:|
> |Baseline($\pi_0$)|1x|102|92|86|
> |V-AEFusion|1.3x|77|88|78|
>
> |Franka|Speedup|Latency (ms)|Cleaning Table (SR%)|
> |:-:|:-:|:-:|:-:|
> |Baseline($\pi_{0.5}$-droid)|1x|95|76|
> |DP-Cache(S=4)|1.28x|74|70|
>
> We will add the real-robot results in the revised version.

---

> > ### Author Rebuttal · Reviewer_ZNEF · 2026-04-02
> >
> > The authors have resolved my concerns and I decide to keep my ratings of weak accept.

---

> > > ### Author Response · Authors · 2026-04-03
> > >
> > > Thank you for the positive feedback. We appreciate your valuable time and effort.

---

### Official Review · Reviewer_eBf2 · 2026-03-24

**Soundness:** 4
**Presentation:** 3
**Significance:** 4
**Originality:** 3
**Overall Recommendation:** 4
**Confidence:** 3

**Summary:**

This study proposes a framework designed to bridge the gap between large-scale VLA (Vision-Language-Action) models and low-cost robotic hardware by characterizing the interplay between software and hardware. The primary contributions of this work include: (1) A systematic benchmarking of VLA pipelines across GPU, NPU, and CPU platforms. To facilitate hardware selection for mobile robots, this study employs a comprehensive metric called CET (Cost, Energy, Time) for evaluation. (2) The introduction of a performance profiling methodology capable of visualizing the "computational seesaw effect" that occurs during VLA execution—a phenomenon wherein the VLM backbone and the Action Expert module alternate between computationally intensive and memory-intensive states. (3) The proposal of a suite of acceleration techniques, named DP-Cache and V-AEFusion, which synchronize these unbalanced execution phases by leveraging temporal redundancy and pipeline parallelism. The resulting hardware-centric insights offer significant practical value for real-world hardware deployment.

**Compliance With Llm Reviewing Policy:**

Affirmed.

**Final Justification:**

The authors have resolved my concerns and I will keep my ratings of weak accept.

**Key Questions For Authors:**

1. In Table 1, the Score for $\pi_0$ is the same 91.5 across all XPU types. What platform is this score assumed?
2. In Table 3, the speedup on the Jetson Thor is the lowest among these three XPU, while it is a newer architecture with high-bandwidth memory. Why is the performance significantly lower than the other?
3. In Figure 7,what is the threshold for "stable" ($L1_{rel}$), stable or dynamic?
4. In Table 4, what is the intuition behind why "Goal" are more sensitive than long-horizon tasks?
5. In V-AEFusion, the first 5 denoising steps are conditioned on "stale features" from the previous VLM cycle. Does this lag lead to overshooting or tracking errors?
6. In Section 5.3.1, V-AEFusion relies on a 1:2 latency ratio, so what if a switch to a heavier VLM backbone?

**Limitations:**

This evaluation primarily focuses on the LIBERO benchmark, with limited exploration of performance in high-speed, unstructured real-world environments. Furthermore, the proposed optimizations appear to be sensitive to specific hardware and software stacks, which may limit their versatility on other XPU architectures. The trade-off between aggressive acceleration and long-term control stability in highly dynamic tasks also requires further long-term research.

**Strengths And Weaknesses:**

•	Soundness: The use of Roofline analysis to identify phase-dependent bottlenecks provides a strong theoretical foundation for the proposed optimizations. The experiments cover a wide range of hardware (NVIDIA, Intel, Huawei), which is rare and valuable.
•	Presentation: The paper is well-structured. Figures and charts clearly show differences and rankings. But a detailed analysis is needed.
•	Significance: The work addresses a critical bottleneck in embodied AI: the gap between high-performance foundation models and the tight energy/latency constraints of edge robotics. It provides a practical guide for deploying generalist robot controllers on affordable platforms.
•	Originality: While caching and pipelining are established techniques, their specific combination and application to the dual-phase nature of VLAs are novel. However, the performance gains vary significantly across architectures.

---

> ### Author Rebuttal · Authors · 2026-03-30
>
> We thank the reviewer for the valuable comments and appreciate the recognition that our Roofline analysis provides "a strong theoretical foundation", our hardware coverage (NVIDIA, Intel, Huawei) is "rare and valuable," and our work addresses "a critical bottleneck in embodied AI." We respond to your comments as follows and sincerely hope our rebuttal properly addresses your concerns.
>
> ***Q1.Table 1: π₀ score is 91.5 across all XPU types. What platform is this assumed on?***
>
> The score reflects the model's capability, which is hardware-independent. For a given VLA model,
> different platforms using the same precision (e.g., BF16) do not affect its capability (task success rate). We will add more explanation in the revised paper.
>
> ***Q2.Why Jetson Thor gets the lowest compilation speedup despite being newer with HBM?***
>
> For NVIDIA devices, compilation speedup (`torch.compile`) primarily comes from kernel fusion and memory accesses reduction. The lowest speedup on Jetson Thor stems from two factors:
> 1. **Bandwidth saturation (primary factor)**: Thor's HBM makes memory access already extremely fast so reducing HBM round-trips saves negligible time. As shown in Figure 6, Thor's high ridge point shifts the VLM from compute-bound (as on RTX 4090) to near the memory-bound boundary, limiting the benefits of kernel fusion and memory accesses reduction.
> 2. **Software maturity**: torch.compile support for Thor's newer architecture is still maturing, leading to suboptimal kernel fusion or fallback to unoptimized paths.
>
> ***Q3.What is the threshold for "stable" $L1\_{rel}$, stable or dynamic?***
>
> The threshold is fixed. As shown in Figure 7, the stable segment spans approximately diffusion steps 20–80, where $L1\_{rel}$ remains consistently low. We determine this interval through offline profiling of the diffusion trajectory before deployment. Developing a dynamic threshold that adapts to diverse workloads and models can be a direction of our future work.
>
> ***Q4.Table 4: Why are "Goal" tasks more sensitive than long-horizon tasks?***
>
> First of all, we apologize for a typo in Table 4: the column labeled "Goal" should be "Object".
>
> Then, we explain "Object" task sensitivity: "Long" involves diverse scenes with distinct objects in ~70% of tasks, whereas "Object" requires distinguishing similar items within a single scene (e.g., identically shaped boxes of cream cheese, chocolate pudding, and butter differing only in color/texture). Because Diffusion Policy lacks VLM semantic recognition and relies on low-resolution (224×224) visual features, it is more sensitive to feature approximations such as caching when fine-grained visual discrimination is required.
>
> ***Q5.V-AEFusion: The first 5 denoising steps use stale features from the previous VLM cycle. Does this cause overshooting or tracking errors?***
>
> It may cause minor performance drops but does not cause overshooting or control instability, as validated across all LIBERO tasks. We also validated this claim with new real-world experiments (π₀ on JAKA S5 dual-arm robot). Stability is supported by:
>
> 1. **High temporal coherence**: Under closed-loop control, the temporal gap between adjacent cycles corresponds to the execution duration of a single action chunk. Within this brief window, the robotic arm's pose changes minimally. The VLM outputs exhibit high layer-wise cosine similarity between adjacent timesteps (Key States: 97.99% ± 0.8%, Value States: 96.91% ± 1.5% across 200 consecutive inference cycles) as shown in https://re-01-figures.vercel.app/ Figure 2, confirming that stale features remain close approximations of the current state.
> 2. **Coarse-to-fine generation**: In diffusion models, early denoising steps establish the rough action trajectory, while the final steps conditioned on fresh VLM features recover precise spatial details. This late-stage refinement corrects minor deviations preventing overshooting.
>
> Our real-world ablation study (https://re-01-figures.vercel.app/) Table 1 confirms this: using up to 5 stale steps yields only a marginal success rate drop (92%→88% and 86%→78%), while reusing more steps causes sharp degradation (50% at 7 steps). This validates our default choice of 5 stale steps as a well-balanced operating point. We will include this ablation in the revised paper.
>
> ***Q6.V-AEFusion relies on a 1:2 VLM:AE latency ratio. What if a switch to a heavier VLM?***
>
> The 1:2 ratio is not a strict requirement for our pipeline. If a heavier VLM shifts the ratio, V-AEFusion still effectively hides the latency of the stale AE steps (e.g., the first 5 steps for π₀). To handle varying ratios, we can adopt a dynamic pipeline boundary: the number of concurrent stale steps is adjusted based on the specific model and workload.
>
> ***W1.Paper is well-structured but "a detailed analysis is needed."***
>
> We will supplement V-AEFusion with real-robot validation and clarify the model scores and threshold for "stable" $L1\_{rel}$ in the revised version.

---

> > ### Author Rebuttal · Reviewer_eBf2 · 2026-04-03
> >
> > The authors have resolved my questions and I keep my ratings of weak accept.

---

> > > ### Author Response · Authors · 2026-04-03
> > >
> > > Thank you for the positive feedback. We appreciate your valuable time and effort.

---

### Decision · Program_Chairs · 2026-04-30

**Decision:**

Accept (regular)

**Comment:**

The paper received uniformly positive reviews: all four reviewers rated it Weak Accept, and after rebuttal, all indicated their concerns were largely resolved and kept their scores. The majority of technical concerns focused on evaluation completeness and deployment realism. E.g., reviewers asked for stronger baselines, clearer guidance on hardware selection using CET, better justification of asynchronous design (V-AEFusion), validation beyond LIBERO, and explicit treatment of memory/VRAM as a hard constraint; one reviewer additionally noted limited theoretical insight/novelty. In rebuttal, the authors addressed these by clarifying the deployment guideline, clarifying baseline choices and trade-offs, adding/committing further experiments (π0 / π0.5, real robot), and explicitly incorporating memory feasibility. Final ratings are still 4 × Weak Accept.

I recommend Accept. Reviewers consistently agreed that the work is technically sound and practically valuable, and concerns are mostly resolved. The AC view is that this is an important problem for VLA deployment, and even if the contribution is largely system-focused, it provides necessary insight and tools for real-world use.